# Electrochemical Immuno- and Aptamer-Based Assays for Bacteria: Pros and Cons over Traditional Detection Schemes

**DOI:** 10.3390/s20195561

**Published:** 2020-09-28

**Authors:** Rimsha Binte Jamal, Stepan Shipovskov, Elena E. Ferapontova

**Affiliations:** Interdisciplinary Nanoscience Center (iNANO), Aarhus University Gustav Wieds Vej 14, DK-8000 Aarhus, Denmark; ribja@inano.au.dk (R.B.J.); stepan.shipovskov@gmail.com (S.S.)

**Keywords:** electrochemistry, bacteria, electrochemical ELISA, electrochemical immunoassays, electrochemical aptamer-based assays

## Abstract

Microbiological safety of the human environment and health needs advanced monitoring tools both for the specific detection of bacteria in complex biological matrices, often in the presence of excessive amounts of other bacterial species, and for bacteria quantification at a single cell level. Here, we discuss the existing electrochemical approaches for bacterial analysis that are based on the biospecific recognition of whole bacterial cells. Perspectives of such assays applications as emergency-use biosensors for quick analysis of trace levels of bacteria by minimally trained personnel are argued.

## 1. Introduction

Sensitive, selective and quickly responding sensors for bacterial detection are strongly required for environmental monitoring of water safety [1,2]; in diary, food and beverage industries for product quality analysis [3]; for prevention, diagnosis and antibiotic treatment of infectious diseases caused by pathogens [4]; and for combatting biocorrosion in oil and gas industries [5], biological warfare and terrorism [6]. Pathogenic bacteria cause more than 10 million deaths annually [4] and 1.5 million people pass away from diarrhea caused by microbiologically contaminated water, with >2 billion people simply lacking access to safe drinking water resources [2]. Microbiological safety requires no bacteria present in any 100 mL of drinking water [7]. Similarly low levels of pathogens should not be surpassed in dairy products (<10 CFU mL^−1^
*E. coli*, a specific indicator of the diary product spoilage, in pasteurised milk) [8], brewery products (<400 CFU L^−1^
*L. brevis* in beer) [9] and food (<100 CFU *Salmonella* gives from 0.01 to 0.56 probabilities of illness; both 17 and 36 CFU can cause illness) [10]. Less than 100 CFU mL^−1^ pathogens should be timely, within 1–2 h detected in blood [11], and unavailability of rapid sensitive tests is partially responsible for high 30–40% mortality from blood stream bacterial infections. Complex matrix effects and bacterial cross-interference further challenge the required analytical sensitivity and limits of detection (LOD).

It is thus no surprise that bacterial sensors should be uniquely robust and specific, with a minimal (if any) false signal to ensure effective protection of human health. Such sensors can be often considered as emergency-use sensors since they should provide quick analysis of trace—“alarm”—levels of bacteria, without complex sample preparation and by minimally trained people. They should be sensitive at a single cell level and trace bacteria immediately in the “alarm” spots.

The Human Microbiome Project further revealed the vital role bacteria play in human health and development of gastrointestinal diseases, several types of cancer and type 2 diabetes [12]. Poorer or uncommon gut microbiota (aka dysbiosis) can also weaken and destabilize immune responses, and its role in development of such neurologic disorders as Alzheimer’s disease, through neuro-inflammation, is more and more acknowledged [13]. Antibiotic resistance is another relatively recently emerged field of bacterial sensor applications: >0.7 million people die annually from drug-resistant infections, and their number is predicted to increase to 10 million in 2050 [14]. These findings highlight the necessity of new, even more advanced analytical tools for multiplex bacterial analysis in complex biological matrix containing excessive amounts of numerous bacterial species.

Here, we overview the existing approaches for bacterial detection, placing the main focus on electrochemical immunoassays and aptamer-based assays for the whole bacterial cells, which currently represent the fastest and most straightforward way of bacterial analysis with a minimal complexity of sample preparation, and, thus, have the largest potential for practical and commercial use. We critically discuss pros and cons of such electrochemical assays compared to traditional immunoassays and their propensity to satisfy the requirements of the ideal bacterial assay’s sensitivity, selectivity and response time.

## 2. Methodological Approaches for Bacterial Detection and Quantification

Most routine “gold standard” microbiological tests for bacterial analysis are microbiological culturing and such nucleic-acid-based approaches as the polymerase chain reaction (PCR) and fluorescence in situ hybridization (FISH) analysis of bacterial DNA and RNA (Figure 1).

Despite the established reputation in the field, some of their inherent limitations preclude their emergency-use applications. Both microbiological culturing and PCR and FISH are time consuming (from 2 to sometimes 15 h of amplification [15] and from 24–72 h to weeks of bacterial growth [16]). In addition, despite their high specificity, they are still insufficiently sensitive [16]. In the fastest PCR and FISH, from 10^3^ to 10^4^ CFU mL^−1^ can be detected [3,16]. Along with that, errors in PCR amplification and sequence replication and differences in the DNA extraction protocols existing for Gram-negative and Gram-positive strains may result in the wrong quantification of bacterial species [17], not to mention the probability of false-positive signals from dead cells, whose DNA can also be PCR-amplified. Amplification inhibition by the matrix components can also contribute to errors in bacteria quantification and therefore detection [18]. Real-time (quantitative) PCR (qPCR) quantifies amplified bacterial DNA in real-time and may be faster/more sensitive than traditional PCR, but requires fluorescent labels and as such a more complex optical detection equipment [19]. It may also suffer from matrix inhibitors and dead-cell produced false positives [20], though some strategies using intercalating dyes eventually allow to prevent amplification of the dead cells’ DNA [21].

The recent progress made in molecular biology tools offered more promising approaches for bacterial nucleic acid (NA) analysis [22] such as:-NA sequence-based amplification (NASBA, amplifies and detects bacterial messenger RNA, more sensitive and fast (less than 90 min) than PCR, no interference from dead cells’ DNA, however, too expensive for environmental applications and suffers from errors in amplification and quantification following the amplification step) [23];-Loop-mediated isothermal amplification (LAMP, less expensive than PCR, more sensitive and faster (1 h) DNA amplification at 60–65 °C, less sensitive to inhibitors) [24];-Recombinase polymerase amplification (RPA, fast (<20 min) amplification of DNA/RNA at 37–42 °C; can be integrated with other, portable detection devices, however, it faces primers design difficulties and requires post-amplification purification digestion) [25].

These new approaches for NA amplification enable the qualitatively different level of analysis, particularly, when combined with a proper detection strategy (Table 1 summarizes some selected best examples). However, despite their current wide use in the clinical and molecular-biology research practice, the NA-based methods, exploiting lab-run equipment and often requiring from 24 to 72 h sample pre-enrichment, have not yet found broad applications in the in-field or alarm situations requiring fast and specific, often single-cell bacterial detection. Analytical platforms such as genosensors [26,27] and DNA microarrays [28,29], or next generation sequencing platforms [30,31] also use bacterial DNA/RNA isolation and amplification protocols as a sample preparation step, and, being more sensitive, they still rely on lab instrumentation and pre-enrichment steps. Centrifugal lab-on-chip microfluidic platforms integrating cell lysis and amplification procedures in one chip are very perspective since they do not need trained personal for handling them and can decrease the time of analysis down to 70 min [32,33]. On the other hand, due to the small, µL-volume samples they use for analysis, the LODs they show are quite high, from 10^3^ to 10^4^ CFU mL^−1^ (down to 10 CFU mL^−1^ when the assay time is increased, e.g., to 3.7 h) [32], making them unsuitable for many applications.

Immunoassays for whole bacterial cells is an alternative strategy intensively used [3,16] (Figure 1). The basic principle of immediate capturing of bacteria by the immobilized highly-specific biorecognition element, such as an antibody (Ab) or an apatmer, followed by a further read-out of the binding event either label-free or through the enzyme-amplified reaction, is most attractive for the development of rapid, sensitive and specific bacterial assays. Immunological analysis of bacteria by, e.g., traditional enzyme-linked immunosorbent assays (ELISA) may be quite specific and eventually fast; however cross-interference and surface fouling in physiological matrices may obstruct the results [3]. Also, the most frequently reported LODs, from 10^4^ to 10^6^ CFU mL^−1^ [16], are insufficient for many emergency-use sensing applications. The required sensitivity and specificity of analysis and higher speed/lower cost can be reached by using a variety of methodologies, amplification strategies and read-out techniques, as well as bio-mimicking bioreceptors and labels (Table 2 and Table 3). The existing and emerging immunoassay approaches for bacterial analysis are scrutinized in the following sections.

Standing separately are metabolic sensors based on the detection of signals associated with specific reactions related to bacterial metabolism [100,101] (Figure 1). Those approaches may be very efficient but generally are either low specific or based on very individual metabolic biomarkers of specific bacterial species, and thus they do not form general analytical platforms.

## 3. Immunoassays and Aptamer-Based Assays with Optical Detection

Immunoassays and aptamer-based assays exploit the specificity of a biorecognition of targeted bacterial pathogens by mono- and polyclonal Abs [102] and by their in vitro alternatives—aptamers [103], and thus rely on their availability. A variety of bacterial pathogens can be detected once Ab or aptamers are developed for a specific bacteria type. The Ab and aptamers also represent the main limitation of this approach, since the available Ab or aptamers may not be sufficiently specific or show insufficient affinities for their targets. They may be too expensive for a number of applications and, finally, they may not be available at all.

In the optical immunoassays for whole bacteria, bacterial binding to the sensor surface changes the optical properties of the sensor/reaction media, such as UV/vis absorption, fluorescence, luminescence, which is optically read out [104]. Ca. 25 and 16 CFU mL^−1^ LODs were shown for *S. typhimurium* and *E. coli* in a 1 h fluorescence assay with aptamer-modified fluorescent-magnetic multifunctional nanoprobes [105]. A 30 min fluorescence aptamer-based assay with vancomycin-Au nanoclusters and aptamer-modified Au nanoparticles (NPs) allowed to detect down to 20 CFU mL^−1^ of *S. aureus* in PBS with no interference from other species; the assay performed well in milk, juice and human serum 10- and 5-fold diluted with PBS [106]. Such assays usually rely on clinical laboratory-based equipment and are less suitable for in-field applications, though on a few occasions portable optical-fiber systems [63,107] or colorimetric lateral-flow tests [64,108] have been reported. Such label-free optical approaches as surface plasmon resonance (SPR) that follow the changes in the refractive index of the transducer bioreceptor-modified interface are generally less sensitive (Table 2), and complex matrix components can strongly interfere with analysis. They also require a bulky equipment low suitable for in-field analysis and point-of-care testing (POCT).

Among optical immunoassays, ELISA is the dominant Ab-based methodology that relies on specific binding of bacteria to Abs immobilized either to a solid support (typically polystyrene, polyvinyl or polypropylene, or in a 96 or 384 micro-well plate) or to magnetic beads [109]. Binding of bacteria by a capture Ab is followed by reactions with a secondary Ab (or an aptamer) typically labelled with such redox enzymes as horseradish peroxidase (HRP) or alkaline phosphatase (AlkP), whose enzymatic reactions with their substrates result in the optically active products. Upon addition of the corresponding enzyme substrates, the reaction mixture changes the color, being followed by the plate reader or other spectrophotometric equipment [110]. HRP and AlkP enzymes are most commonly used in the enzyme-dependent immunoassays, both due to their high turnover rates and satisfactory stability of their bio-conjugates at 4 °C (both are sensitive to freezing [111,112,113]). In situations of a high-level endogenous peroxidase activity, AlkP becomes a more suitable choice. Among other frequently used enzymes is β-D-galactosidase (β-gal), whose bioconjugates are much more stable and can be stored for at least one year at 4 °C [114]. However, due to its high molecular weight and low turnover number, β-gal is used less. Still, stability and sensitivity issues trigger further search of novel and advanced enzymatic labels, such as adenosine deaminase (ADA) [115], or enzyme-loaded nanostructured labels, such as HRP-loaded nano-spherical poly(acrylic acid) brushes increasing the sensitivity of conventional ELISA by 267-fold [116].

Direct ELISA, in which an antigen-coated micro-well plate is exposed directly to an enzyme-linked Ab [117] is well suited for bacterial analysis [110]. However, the reported LODs of direct ELISA (over 10^6^ CFU mL^−1^ [16,118]) are insufficient for earlier discussed applications. Down to 10^3^ CFU mL^−1^ LODs can be reached with a sandwich ELISA, in which the signal is generated only when a complete primary Ab(bacteria)secondary Ab sandwich is assembled, and in indirect ELISA (Table 2). Such setups are reported to be 2–5 times more sensitive than direct and other ELISA types [119]. 10^3^ CFU mL^−1^ of food-borne pathogenic *Salmonella* could be detected in indirect ELISA by targeting its membrane protein, bacterial flagellin FliC [120], while whole bacterial cell sandwich ELISA allowed detecting 10^3^ CFU mL^−1^ of *Brucella abortus* and *Brucella melitensis* [121] and *Bacillus cerus* [122]. Further specificity and sensitivity of analysis can be controlled by a proper choice of Abs and Ab-replacing aptamers and of aptamer-reporter labels and nanomaterial-based labels [123]. Another emerging development of ELISA is a replacement of the primary Ab by bacteriophages, which allowed specific detection of *E. coli* and *Salmonella* strains, with a LOD of 10^5^ cells per well (or 10^6^ cells mL^−1^) [124]. Overall, the shown LOD and eventually the protocols requiring lab-operating equipment make traditional ELISA low suitable for sensitive emergency-use applications.

Lateral flow immunoassays (LFI) share with ELISA the basic immunological principle and offer advantage in cost and faster analysis times. Adapted for a dipstick or immunochromatographic strip operation, they can essentially simplify and accelerate the analysis of the pathogenic microorganisms, since they do not require complex equipment or special training for their handling, and thus are more suitable for POCT [125] (Table 2). Colorimetric lateral-flow tests have reported LODs varying between 100 CFU mL^−1^ (a 5 min sandwich assay for *E. coli* with an HRP label read out by a CCD camera) [108] and 10^4–^10^5^ CFU mL^−1^ (a 20 min LFI with up-converting phosphor NPs as reporters, 10 CFU 0.6 mg^−1^ after pre-enrichment step) [126]. Similarly, 10 CFU mL^−1^ of different bacterial species could be detected in 1 h only after from 4 to 5 h pre-enrichment steps [127]. It is clear that without pre-enrichment most of LFIs show sufficiently high LODs (from 10^2^ to 10^6^ CFU mL^−1^) [128,129,130], which currently restricts their immediate application in “alarm” situations.

## 4. Electrochemical Immunoassays

Electrochemical biosensors for bacteriological analysis include: (A) genosensors for bacterial DNA or RNA; (B) bacterial metabolic sensors, and (C) biosensors for detection and quantification of the whole bacterial cells [131]. Of those, electrochemical immunoassays and aptamer-based assays for bacteria are most challenging as platform biotechnologies competing with commercially established optical immunoassays. Due to a portable and inexpensive equipment with minimal power requirements and more robust read-out techniques, electrochemical approaches often allow more rapid and accurate bacterial detection with a decreased cost of equipment/assay and easier miniaturization of the device for use in-field and at POCT sites.

In the electrochemical approach, binding of bacteria to the Ab- or aptamer-modified surfaces is detected electrochemically by means of a redox active indicator (also: a redox mediator and a redox active product of the enzymatic reaction) or label-free, through the changes in the interfacial properties of the modified electrodes (Figure 2A–C). Overall, the specificity and sensitivity of the bacterial immunoassays can be radically improved by a wise combination of bio-recognition abilities of Abs/aptamers and redox indicators with electrochemical methodologies [27,123]. In electrochemical sandwich ELISA (e-ELISA) the formation of the Ab(bacterium)Ab complex is detected electrochemically by recording the electro-enzymatic activity of the labels, such as HRP or AlkP. The signal is amplified as a result of electrochemical recycling of one of the enzyme substrates at the electrodes or due to accumulation of the product changing the electrode properties (Figure 2D–F). The electrochemical signals can be further enhanced by modifying the electrode surface and/or designing more sophisticated capturing and detection mechanisms (Table 3).

In contrast to optical immunoassays, electrochemical immunoassays (e-immunoassays) for bacteria may not need any labeling at all since they can rely on the interfacial changes resulting from the bacterial binding to the bioreceptor-modified surface. The microscopic, from 0.5 to 5 µm size of bacteria results in significant changes of the electrical properties of the bio-recognition interface upon bacterial binding, although surface fouling by non-specifically bound bacterial species may be a serious electroanalytical issue. The sensitivity and specificity of e-immunoassays is therewith essentially increased compared to traditional ones, the LOD being improved to just a few CFU [131,132]. The most sensitive and fastest appeared to be impedimetric biosensors and e-immunoassays on magnetic beads (MBs) enabling a few CFU mL^−1^ bacteria detection (Table 3). In addition, both the sample volume and the time of analysis can be significantly minimized, which is important for emergency applications. Detection protocols can be adapted for POCT (particularly, within the microfluidic format [98]) or portable-device in-field operation.

### 4.1. Electrochemical ELISA

Most intensively reported is e-ELISA exploiting HRP or AlkP bioelectrocatalytic labels, whose substrates are either electrochemically recycled at electrodes [77,133,134] or precipitate and block the redox indicator reactions at the electrode [135] (Figure 2D). Using traditional ELISA’s enzymatic labels that rely on their substrate recycling at electrodes may result in a strong dependence of the e-ELISA sensitivity on the electrode surface properties. It often results in LODs close to those reported in the optical ELISA schemes. Bioelectrocatalytically-amplified ELISA on MBs is more advanced (Figure 2E). Bacteria are collected on MBs modified with bacteria-specific Ab (or aptamers), immunomagnetically separated from the original, often complex bacterial sample matrices, and finally pre-concentrated in smaller volume samples [136]. That results in sample amplification and excludes biofouling of the electrodes with matrix components. 845 CFU mL^−1^ of *S. aureus* in nasal flora samples could be specifically detected in the 4.5 h sandwich e-ELISA on MB with the AlkP label [137]. A competitive e-immunoassay with the HRP label allowed detecting 1 CFU mL^−1^
*S. aureus* in 2 h and 1.4 CFU mL^−1^
*Salmonella* in 50 min, in milk, without any sample pre-enrichment and with exceptional selectivity over other pathogens [77].

Requirements for a higher stability, simplicity and lower cost of e-immunoassays triggers application of low-cost redox inactive hydrolases as enzymatic labels in e-ELISA: lipase [138], urease [89] and cellulase [139] can also be effectively used in bacterial RNA and protein e-ELISA [9,140,141]. With hydrolase labels, products of hydrolysis of their substrates are electrochemically detected: bio-transformation of urea to ammonium carbonate increases the impedance of the system, while cellulase digestion of nitrocellulose films formed on graphite electrodes increases their electronic conductivity (Figure 2F). Such assays by itself rely not on the reactivity of the electrode, but rather on the enzymatic label reactivity with their substrates. That allowed highly sensitive and specific detection of 12 CFU mL^−1^ of *E. coli* in the buffer solution and in milk, in 2 h [89], and down to 1 CFU mL^−1^ of *E. coli* in tap water (2 CFU mL^−1^ of *E. coli* in milk) within 3 h, by assembling a hybrid aptamer (*E. coli*)Ab sandwich on MBs [90]. 100 CFU mL^−1^ of *Salmonella enteretidis*, *Enterobacter agglomerans, Pseudomonas putida*, *Staphylococcus aureus* and *Bacillus subtilis* did not interfere with the single *E. coli* detection when Ab was used as a capture element [90]. Both assays did not require any sample pre-treatment (e.g., cell pre-enrichment), are electrochemically label-free (no redox indicator/mediator was used), and are cost-effective due to the low cost of urease ($0.5 per mg) and cellulase (from $0.002 to $0.2 per mg) and their high storage stability. The urease assay could be well-integrated within the microfluidic format [89]. Both assays are general and can be adapted for specific detection of any other bacterial species once the corresponding Ab and aptamers are used.

Efforts are also focused on replacing enzymatic labels by different type of nanomaterial-based labels and catalysts such as quantum dots (QDs), DNAzymes and electrocatalytic nanoparticles (Table 3). For example, 3 CFU mL^−1^ of *E. coli* O157:H7 was detected in spiked milk samples by a sandwich e-ELISA assembled on the Ab-poly(*p*-aminobenzoic acid)-modified electrode and labelled with CdS QDs encapsulated in a metal organic/zeolitic imidazolate framework-8 (CdS@ZIF-8). The large load of the metal framework with CdS QDs resulted in an amplified electrochemical response detected voltammetrically [142].

The most successful example is the traditional HRP-linked sandwich ELISA integrated in the automated microfluidic electrochemical device, in which the HRP-labelled Ab was replaced by the HRP-Ab-Au NPs complex [98]. 3,3′,5,5′-Tetramethylbenzidine was an electrochemical mediator of the H_2_O_2_ reduction by HRP detected chronoamperometrically at −0.1 V. Sandwich assembly on the gold microelectronic chip surface then allowed down to 50 CFU mL^−1^ detection of *E. coli* in water within the overall 20 min procedure, with no interference from *Shigella*, *Salmonella* spp., *Salmonella typhimurium*, and *S. aureus*. It was possible to regenerate the Ab-modified sensor surface in a flow of 0.1 M HCl and then re-use it for another *E. coli* detections [98]. In a microfluidic assay, the fast delivery of the bacteria to the sensor surface by the microfluidic flow is equivalent to the fast capturing of bacteria on MBs. However, due to small volumes of injected samples (e.g., 200 µL injected for 8 min at 25 µL min^−1^) [98] and turnover limitations of redox enzymes used, a 1 CFU mL^−1^ LOD may be difficult to achieve.

### 4.2. Electrochemical Immunoassays (Not Enzyme-Linked)

#### 4.2.1. Antibodies and Aptamers Based Assays

The simplest e-immunoassay strategy represents the electrode surface modification with an Ab or an aptamer, and immediate electrochemical detection of bacterial binding, either by following the interfacial changes accompanying binding impedimetrically, with or without a redox indicator, or by electrochemical monitoring of the metabolic activity of the captured cells (Figure 2A–C). The latter allows assessment of a viable cell population. *E. coli* and *N. gonorrheae* were immuno-specifically captured on the Ab-modified gold screen-printed electrodes (SPE), and 10^6^ and 10^7^ CFU mL^−1^ of them were quantified by voltammmetric analysis of the electroenzymatic activity of bacterial cytochrome *c* oxidase; synthetic substrate *N*,*N*,*N*′,*N*′-tetramethyl-*p*-phenylene-diamine was used as a mediator [143] (Figure 3A). The assay was relatively fast (45 min binding and 1 h SPE regeneration) and inexpensive. Since the LOD was quite high, the authors suggested it for assessment of the efficiency of antibiotic treatments since it relies on the assay ability of a viable cells quantification.

The most frequently used e-immunoassays detect the total number of bacterial cells bound to the Ab or aptamer-modified electrodes by using Electrochemical Impedance Spectroscopy (EIS) either in the presence of a redox indicator, such as ferricyanide, or without it [84,148,149,150,151] (Table 3). Significant changes in the interfacial properties of the Ab-modified electrodes after binding of bacterial cells allowed 1 h detection of 10 CFU mL^−1^ of *E. coli* CIP 76.24 strain (Ab immobilization on biotinylated alkanethiol SAMs) [84] and 5.5 CFU mL^−1^ of *Listeria monocytogenes* (Ab immobilisation via protein A capable of binding of the Fc-region of Abs) [149].

10^4^ CFU mL^−1^ of *S. pyogenes* was detected in human saliva in 30 min by bacteria capturing at the Ab/biotinylated Ab-modified electrodes (immobilization on NHS/EDC-activated alkanethiol SAMs or a conductive polymer via biotin-streptavidin/neutravidin linkage) [148]. 100 CFU mL^−1^ of *E. coli* O157:H7 were detected at Ab-modified screen-printed interdigitated microelectrodes (immobilisation through the 2-dithiobis(sulfosuccinimidylpropionate)) in less than 1 h, and wheat germ agglutinin capable of binding to bacterial cell walls was used for the EIS signal enhancement [151]. Only 10 min took 600 CFU mL^−1^
*S. eneteritidis* detection at the aptamer/Au NPs-modified carbon SPE [150]. Aptamer immobilization on the Au NPs-modified electrodes was performed through the alkanethiol linker introduced in the 5′ end of the aptamer sequence.

Microfluidic formats of the impedance immunoassays further improve the assay time: 7 CFU mL^−1^ of multiple *Salmonella* serogroups were detected in 40 min in poultry and lettuce samples with no interference from *E. coli*, by bacteria trapping at the interdigitated Ab-modified Au electrodes of a microfluidic biosensor [152]. In the latter case, live and dead cells were discriminated by their different intensity of the impedance signal. However, no signal calibration data or cell growth information were given.

Generally, e-immunoassays relying on the surface-immobilized Ab and aptamers are quite specific, though, in many cases the modified electrodes also respond to other bacterial species, yet with a signal amplitude less significant than with the targeted bacteria [84,148,150]. To prevent non-specific binding of other bacteria, such antifouling strategies as electrode patterning and co-adsorption of antifouling agents were used. 100 CFU mL^−1^ of *E. coli* O157:H7 with no interference from *E. coli* K12, *Salmonella typhimurium*, or *S. aureus* were detected at the aptamer-modified three-dimensional interdigitated microelectrodes (3 µm in width and 4 µm in height) separated by insulating layers [153]. Antifouling properties of polyethylene glycol (PEG) in combination with its ability to inhibit the electrode reactions of ferri/ferrocyanide [154] were used to detect down to 10 CFU mL^−1^ of uropathogenic *E. coli* UTI89 in serum and urine at the Ab/reduced graphene oxide/polyethylenimine (PEI)-modified electrodes (Ab immobilization via formation of amide bonds with PEI and further electrode modification with pyrene-PEG) [155]. The biosensor discriminated wild type *E. coli* UTI89 from UTI89 Δfim strain.

#### 4.2.2. Antimicrobial Peptides (AMP) Based Assays

AMP may be considered as peptide aptamer alternatives to Ab and oligonucleotide-based apatmers [156]. Though their specificity for different bacterial cell strains may be lower, they can provide the basis for broader platforms for the pathogen detection. AMP-modified micro-fabricated interdigitated electrodes allowed 12–15 min detection of down to 10^3^ CFU mL^−1^ of *E. coli* and *Salmonella* [157]. AMP magainin I (GIGKFLHSAGKFGKAFVGEIMKS) bearing the net positive charge was immobilized on gold interdigitated microelectrodes via the extra Cys residue introduced in its C terminus and recognized a number of heat-killed pathogenic bacterial strains such as *E. coli* O157:H7 and *Salmonella*. AMP was concluded to primarily interact with the negatively charged phospholipids of Gram-negative bacterial membranes through the positively charged amino-acids in its N-terminal region. Also, the insignificant affinity was shown for Gram-positive (lacking the phospholipid-containing outer membrane) and non-pathogenic *E. coli* species (their cell walls miss hydrophilic O antigens essential for electrostatic and hydrogen bonding).

#### 4.2.3. Bacteriophages Based Assays

Bacteriophages (or simply phages) are another perspective bio-recognition element alternative to Ab. Those are chemically and thermally stable viral nanoparticles capable of specific interactions with host bacteria and their infection. Phages’ surface peptides display the aptamer properties towards bacterial surface proteins, and these properties can be modulated and optimized both chemically and genetically. Currently, phages are intensively explored in bacterial e-immunoassays as bio-recognition capturing probes enabling not only specific binding but also discrimination between viable and dead cells [158].

For intact bacterial sensing, phages are immobilized on electrodes by either physical adsorption, resulting in random surface orientation of phages, or directed orientation approaches (Figure 3B). Adsorption of *E. coli*-specific T4-phages on gold-nanorods-modified pencil graphite electrodes produced a biorecognition layer able of EIS detection of 10^3^ CFU mL^−1^ of *E. coli* cells (no interference from *S. aureus*) [159]. The EIS responses strongly depended on the reaction time, being maximal after 25–35 min of *E. coli* binding, and then dropped down because of bacterial lysis by the phage. To improve the binding affinity of the phage SAMs, oriented T4-phage immobilization by its covalent binding either to cysteamine or already activated 3,3′-dithiodipropionic acid di(N-hydroxysuccinimide) ester was combined with the alternating electric field-modulation of the phage orientation. Such electrode polarization increased the number of phages properly oriented for *E. coli* binding, which resulted in the improved 100 CFU mL^−1^ LOD after 15 min binding reaction [160]. A very similar quantification of *Salmonella* after 50 min of bacteria binding reaction was reported for the capacitive flow system with polytyramine-modified gold electrodes modified with the covalently attached M13 phage specific for *Salmonella* spp. [161] (Figure 3B). The sensor surface could be regenerated 40 times by the alkaline treatment.

The cell-lytic properties of phages may interfere with reaching low LOD in bacterial analysis due to the fast lysis of infected bacterial cells. The sensitivity of bacterial analysis can be improved by using non-lytic phages, such as a non-lytic M13 phage that could recognize F+ pili of *E. coli* XL1-Blue and K12 strains: it was covalently attached to 3-mercaptopropionic acid-modified AuNPs via EDC/NHS chemistry [88]. That allowed increasing the time of the reaction between the immobilized phage and *E. coli* cells, and 14 CFU mL^−1^ of *E. coli* was detected by EIS. Still, the best LODs obtained with AMPs and phages do not approach those observed in the Ab- and aptamer- based e-immunoassays.

Along with that, lysis itself can be analytically useful. 10^3^ CFU mL^−1^ of *E. coli* B were detected by EIS, with no interference from the K strain, at the T2 phage-modified electrodes [144]. A T2 phage specific for *E. coli B* strain was immobilized through its negatively charged head on the PEI/carbon nanotubes–modified glassy carbon electrode positively polarized. Such polarization-directed immobilization of the phage enabled selective binding of the targeted *E. coli* cells for a time sufficient for bacterial cell infection and lysis by the phage that was impedimetrically detected.

### 4.3. Whole Cell Imprinted Polymer Sensors as Alternative to E-Immunoassays

Cell-imprinted polymers (CIP) are biomimetic synthetic Ab alternatives for e-immunoassays. Turner’s group produced a CIP sensor based on electropolymerized 3-aminophenylboronic acid (3-APBA) [162]. Polymerization of 3-APBA monomers lead to the formation of the *cis*-diol-boronic group complex within the template matrix that facilitated reversible binding and easy release of the trapped bacterial cells (upon subsequent regeneration) from the CIP. EIS responses of the CIP sensor were proportional to log 10^3^–10^7^ CFU mL^−1^ of *Staphylococcus epidermidis*, and the sensor did not respond to other similar shape bacteria species [162]. Impedimetric analysis with ferricyanide as a redox indicator detected less than 1 CFU mL^−1^ of uropathogenic *E. coli* UTI89 bound to ultrathin silica films prepared by sol-gel technology on gold-coated glass slides into which *E. coli* was imprinted (signal linearity from 1 to 10^4^ CFU mL^−1^), with *S. aureus* and *Pseudomonas aeruginosa* as negative controls [145] (Figure 3C). No information about the time of the assay or analyzed sample volumes versus the electrode size were reported for this very impressive assay, though. Thus, CIP can be an inexpensive replacement for the biological recognition elements if produced/shown to be sufficiently specific. Along with that, CIP may not provide the necessary specificity to discriminate between the different strains of the same bacteria, since they rely mostly on the bacterial shape and size and less on the bacterial surface peculiarities.

### 4.4. Electro-Optical Immunoassays

Combination of electrochemistry with an optical readout can further generate new POCT devices with improved sensitivity for bacterial analysis. Abs for *E. coli* were coupled to films of electropolymerized polyaniline (PANI) on ITO screen-printed electrodes, whose polarization changed the PANI oxidation states and generated concomitant changes in the film color different for *E. coli* bound and unbound films [146] (Figure 3D). Different electrochromic responses due to the presence of *E. coli* increasing the interfacial resistance and thus affecting the PANI oxidation states, allowed to detect down to 10^2^ CFU mL^−1^ of *E. coli* by the naked eye, while 10 CFU mL^−1^ could be detected by a software.

Electrochemiluminescence (ECL) is another electro-optical approach that improves immunoassay’s sensitivity by electrogenerated chemiluminescent signal amplification. Most convenient are sandwich immunoassays with a secondary Ab labelled with Ru(II) tris(bipyridine) complex emitting light after electrochemical stimulation with such co-reactant as tripropylamine [163,164] (analogues of e-ELISA in which enzyme labels are replaced by a chemiluminescent reagent). 45 CFU mL^−1^ of *Francisella tularensis* with Ab fragments as capture biomolecules could be detected in a fluidic chip with screen-printed 42 Au electrodes array within a 30 min procedure [164], and down to 2.3 CFU mL^−1^ of *E. coli* O157:H7 were detected in 2 h with the automated electroluminescent 48-well singleplex plate sensor (250 µL samples) [163]. Additional 4 h sample pre-enrichment by ultrafiltration of 10 mL samples further decreased the LOD to 0.12 CFU mL^−1^ [163]. Labelling of the secondary (reporter) Ab with graphene oxide (GO) nanosheets forming multi-complex with Ru(bpy)_2_(phen-5-NH_2_)^2+^ allowed a 1 CFU mL^−1^ analysis of *Vibrio vulnificus* in an overall 2.2 h assay [165]. This approach was referred to as a Faraday-cage type, since the extended GO network enhanced the electron transfer exchange at the electrodes and the intermolecular ECL efficiency by extending the electrode reaction zone.

The ECL amplification can also improve the outcome of the CIP-based e-immunoassays. *E. coli* 0157:H7 was imprinted into the polydopamine matrix by copolymerization, and further binding of *E. coli* was followed from the ECL signals from the next-step bound *E. coli* Abs labelled with N-doped graphene quantum nanodots (in reaction with potassium persulfate) [166]. From 10 to 10^7^ CFU mL^−1^ were detected, with a LOD of 8 CFU mL^−1^.

Another type of ECL immunoassays for bacterial cells exploits the ECL signal inhibition resulting from the electrode surface blockage with bacterial cells captured on the bioreceptor-modified surface [167] (a principle similar to some already discussed e-immunoassays [84,148,149,150,151]). Binding of *E. coli* to the aptamer-modified 3D N-doped high-surface-area graphene hydrogel was detected by following the inhibition of the ECL signal from luminol (in the presence of H_2_O_2_) [167]. AgBr nanoparticles were used as a catalyst for enhancing the ECL of luminol. Down to 0.5 CFU mL^−1^ were detected in spiked buffer solutions after the 40 min *E. coli* binding reaction. No information was provided on the *E. coli* strain or samples composition/volume, though. In another luminol-linked assays, the PaP1 phage specific for *Pseudomonas aeruginosa* was covalently coupled to carboxylated graphene casted on a glassy carbon electrode [168]. Down to 56 CFU mL^−1^ of *P. aeruginosa* was analyzed by following the ECL signal from luminol that decreased after binding of the bacteria [168]. *P. aeruginosa* was quantified in milk and human urine in a 30 min assay.

Despite these impressive results, the adaptation of ECL analysis of bacteria for POCT or in-field analysis seems to be not straightforward. Similarly to optical ELISA, it needs a quite complex read-out equipment. The existing commercial ECL-enabling analyzers, such as Roche cobas^®^ 6000 analyzers, allow from 170 to 2170 test per h, but they are not yet adapted for bacterial sensing [169] and may be not suitable for direct analysis in the blood. More portable devices are nevertheless being developed [164], though their sensitivity should be further improved.

### 4.5. Electrochemical Immunoanalysis of Whole Cells with A Nanopore Technology

Perspective adaptations of the nanopore technology to whole cells electrochemical immunoanalysis are based either on highly specific binding of bacterial cells to Ab [147] or on less specific binding to an AMP [170] immobilized in the nanochannels of a porous alumina or silicon membrane. With this, bacterial binding results in the nanochannels being partially blocked for the ion fluxes. In the first case, *E. coli* binding blocked the nanochannels for the redox indicator reaction at 10 CFU mL^−1^; the viability of cells was accessed with the same redox probe [147] (Figure 3E). In the second case, bacterial outer membrane liposaccharides were recognized by the AMP [170]. The binding affected the diffusivity of the redox indicator within the nanochannels and resulted in the drop of the voltammetric signal. Though no bacterial species were analyzed in this case, the assay was claimed to be generally applicable for any Gram-negative bacteria detection [170].

Both strategies can be eventually adapted for the fast bacterial detection in easy-to-handle microfluidic devices. The Ab–modified nano-porous alumina membrane integrated within the microfluidic chip allowed the simultaneous, from 10^2^ to 10^5^ CFU mL^−1^, impedimetric detection of *E. coli* 0157:H7 and *S. aureus* within ca. 40–50 min (30 min of incubation with bacteria, then washing with a buffer solution, and execution of EIS analysis) [99] (Figure 3F). To prevent non-specific bacterial adhesion, the internal surfaces of the device and the membrane were modified with PEG as an anti-biofouling agent. However, despite the simplicity of the detection scheme and the overall set-up design, the LOD of 100 CFU mL^−1^ seems to be too high for some immediate applications such as water quality analysis (according to WHO, no *E. coli* cell should be present in any 100 mL of drinking water).

## 5. Future Perspectives

Bacterial detection is a dynamically developing field. Emerging new methods for rapid and specific analysis of bacterial pathogens are already contributing to improving our life quality by monitoring health risk situations and decreasing the incidences of illness. For example, during the last decade, confirmed cases of *Salmonella*-caused illnesses have dropped by ca. 40%, due to significantly improved quality of food analysis, with cases reported increased six-fold [171]. Along with that, due to some inherent limitations regarding the bacterial assays’ time, sensitivity and often unaffordable for some application cost, some human activity fields still challenge the bacterial sensor market.

There are complex analytical problems not solved yet in infection disease diagnostics and environmental analysis, such as both fast and specific detection and quantification of a low number of viable pathogens in clinical analysis of blood stream infections, or ultrasensitive, fast and inexpensive water quality analysis in large-volume samples in the presence of excessive amounts of other bacterial species. Fast, 1–5 min assaying of bacteria in inexpensive paper microfluidic [68] and LFI [108] devices is extremely attractive for environmental analysis, but 100–300 CFU mL^−1^ LODs reported make those immunosensors less suitable for ultrasensitive bacterial detection. In clinical analysis, from 1 to 100 CFU mL^−1^ of pathogenic species should be detected rapidly in blood samples for timely diagnosis of bloodstream infections [11], and within several hours antibiotic susceptibility testing should be performed. Despite the recent achievements, state-of-the art microfluidic systems for bacterial analysis in the blood, with their 10^3^ CFU mL^−1^ LOD, are not suitable for practical applications yet [172], and current clinical analysis is still based on microbial culturing coupled with susceptibility tests; both may last for several days to result.

Considering the recent reports discussed, electrochemical immunosensors for whole bacterial cells can undoubtedly solve these problems, at low cost and by a constructive detector and instrumental design friendly to minimally trained personnel. Impedimetric, label-free 40 min e-immunoassaying of 7 CFU mL^−1^ [152] and 20 min e-ELISA of 50 CFU mL^−1^ of *E. coli* [98] in microfluidic devices are promising examples of the devices for real worlds sample analysis. The same refers to the urease-linked e-ELISA on MBs (12 CFU mL^−1^ 2 h) [89] and cellulase-linked e-ELISA on MBs (1 CFU mL^−1^ in 3 h) [90]. Both can be adapted to microfludics and electrochemical LFI, whose electrochemical adaptations are still scarce. Development of cheap electrochemical paper sensors [173,174] is another emerging biosensor trend, and their combination with electrochemical immunomagnetic and phage-based assays can deliver attractive practical solutions for cost-effective and efficient in-field/out-of-lab and POC testing systems.

However, compared to the number of excellent publications, the number of commercialized electrochemical immunosensors biosensors for bacterial pathogens is small. Development and validation of bioelectronic sensor devices capable of efficient solving the real world analytical tasks seems to be slow. Along with that, in addition to electrochemical Accu-Check (Roche Diagnostics, Basel, Switzerland) and Free Style (Abbott Diabetes Care, Chicago, IL, USA) dominating the glucose biosensor market today, Abbott Inc. introduced the e-ELISA platform iStat Systems for blood-circulating protein biomarkers of acute diseases, which becomes a breakthrough in the field of biosensors [175]. With that, e-immunoassays start to slowly crowd the optical ELISA market and may be one day will force it out with advanced electrochemical solutions addressing most urgent problems in the bacterial analysis field.

## Figures and Tables

**Figure 1 sensors-20-05561-f001:**
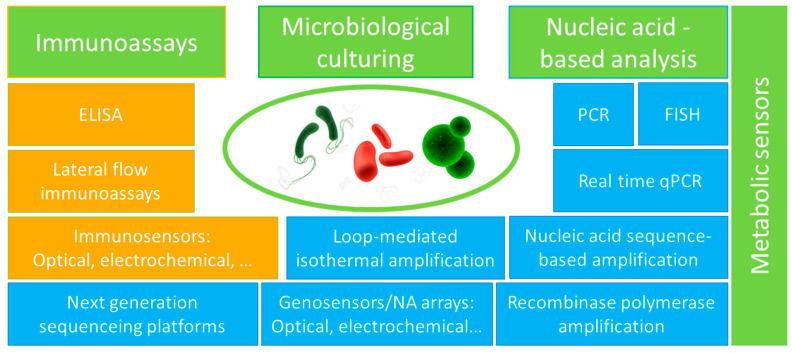
Existing methodologies for bacterial analysis.

**Figure 2 sensors-20-05561-f002:**
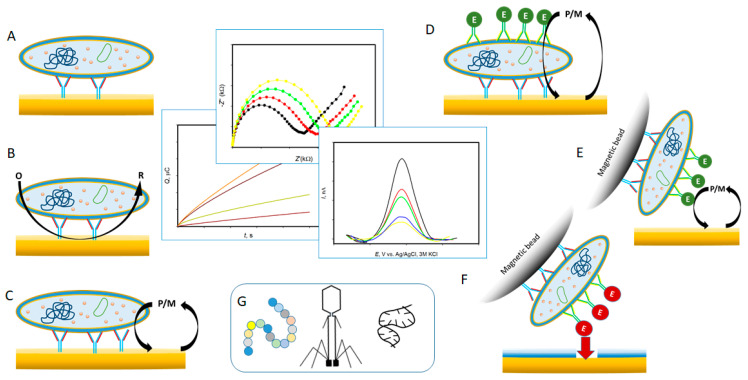
Schematic representation of the typical electrochemical immunosensors. (**A**) In a label-free and indicator-free immunoassay binding of a cell to the Ab-modified surface is detected impedimetrically; (**B**) In the presence of a redox indicator cell binding can also be detected voltammetrically or by chronocoulometry. In (**C**) a redox mediator of cellular metabolism is recycled at the electrode, giving rise to signals associated only with live cells. In (**D**) a bacterial cell is entrapped in an immune-sandwich formed by two Abs on the electrodes surface and labeled with a redox active enzymatic label, whose activity is electrochemically monitored through its substrate recycling at the electrode surface. (**E**,**F**) represent the sandwich assay adaptation to the magnetic beads-format, in (**F**) redox-inactive enzymatic labels induce changes at the electrode-solution interface that are electrochemically detected. (**G**) In (**A–F**) designs elements alternative to Ab can be used: peptides, phages and aptamers.

**Figure 3 sensors-20-05561-f003:**
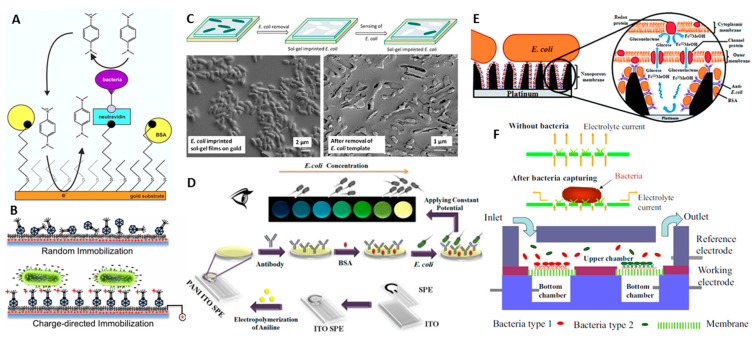
Selected examples of bacterial sensors based on (**A**) immune-recognition and electrochemical assessment of viable cell metabolism [143]; (**B**) Phage-based bacterial cell assay [144]; (**C**) Whole cell imprinted polymer sensor based on *E. coli* imprinting into ultrathin silica films on gold-coated glass slides [145], (**D**) Electro-chromic immunoassay [146]; and (**E**) Electrochemical nanopore immunoassay [147], and (**F**) Nanopore e-immunoassay integrated within the microfludic device [99]. Copyright (2019), (2017), (2019), and (2011) American Chemical Society and copyright (2016) and (2019) Elsevier.

**Table 1 sensors-20-05561-t001:** Selected examples of nucleic acid-based optical biosensors for bacteria.

Strain	Technique	LOD ^a^	Detection Range, CFU mL^−1^	Interference Studies	Assay Time	Ref.
*Campylobacter jejuni*	PCR with pre-enrichment in suitable broths	3 CFU/100 mL	-	*Actinomyces pyogenes, Campylobacter coli, Enterobacter cloacae, Pseudomonas aeruginosa, Salmonella saintpaul, Yersinia enterocolitica*	48 h	[34]
*Salmonella, Listeria monocytogene*	qPCR with two fluorescently labelled primers	5 CFU/25 mL	-	*B. cerus, Campylobacter, E. aerogenes, E. cowanii, Cronobacter sakazakii, E. coli, E. faecalis, S. aureus, Shigella spp, Serratia liquefaciens, S. pneumoniae*	<48 h	[35]
*Salmonella*	q-PCR with two-step pre-filtration on filter paper	7.5 CFU/100 mL	-	-	3 h	[36]
*Salmonella typhimurium*	direct PCR with immunomagnetic preconcentration	2–3	6–6.4 × 10^4^	-	<3 h	[37]
*Salmonella*	PMA-qPCR	36 (pure culture) and 100 (raw shrimp)	36–3.6 × 10^8^ (pure culture) and 100–1 × 10^8^ (raw shrimp)	*Vibrio parahaemolyticus, Listeria monocytogenes, E. coli O157:H7, S. aureus*	1–2 h	[38]
	PMA was used to increase sensitivity					
*Salmonella* sp.	Multiplex qPCR with immunomagnetic pre-concentration	2 CFU/g	-	*Listeria monocytogenes, E. coli, B. cerus, Streptomyces griseus, Pseudomonus aerug, Lactobacillus plantarum, E. faecalis, Streptococcus hemolyticus, Micrococcus luteus, P. aeruginosa, Clostridium sporogenes*	<8 h	[39]
*Shigella* sp.		6.8 CFU/g				
*Staphylococcus aureus*		9.6 CFU/g				
*Salmonella*	Real-time RPA	10 CFU/g (eggs)	-	*Bacillus cereus, Campylobacter coli, E. coli O157:H7, L. casei, S. aureus, Pseudomonas aeruginosa, Vibrio vulnificus*	10 min	[40]
		100 CFU/g (chicken)				
*Salmonella*	LAMP	4.1	-	*Listeria monocytogenes, E. coli O157:H7, S. aureus, Yersinia enterocolitica, Proteus mirabilis, Shigella Flexner, Micrococcus luteus, Bacillus cereus, Enterobacter sakazakii, Pseudomonas fluorescens*	1 h	[41]
*Vibrio parahaemolyticus*	LAMP	530	-	*Acinetobacter baumannii, Aeromonas hydrophila, Enterococcus faecalis, Haemophilus influenzae, Helicobacter pylori, Salmonella*	1 h	[42]
*Salmonella and Shigella*	Multiplexed LAMP Simultaneous detection of two bacterial species	5 CFU/10 mL	-	*S. aureus, E. coli, Bacillus cereus, Pseudomonas aeruginosa, Vibrio parahaemolyticus, Listeria monocytogenes*	20 h	[43]
*E. coli*	NASBA	40	-	*Listeria monocytogenes, Shigella sonnei, Yersinia entero Colitica, Salmonella typhimurium*	40 min	[44]
*Staphylococcus aureus*	NASBA	1–10	-	*Lactococcus lactis, Bacillus cereus, Listeria monocytogenes, Enterococcus faecalis, E. coli, Citrobacter freundii, Sal-monella, Streptococcus bovis, Klebsiella aerogenes*	3–4 h	[45]
*Salmonella enteritidis*.	FRET with CNP for signal enhancement	150	100–3000	*Salmonella typhimurium, E. coli K88*	2 h	[46]
*Salmonella*	DNA Micro-array	2–8 CFU/g (tomato)	-	*E. coli, Shigella, S. aureus, Pseudomonas aeruginosa, Citrobacter freundii, Vibrio cholera, Enterococcus fae-calis, Yersinia enterocolitica*	<2 h	[47]
*Salmonella*	DNA Micro-array QD used in-place of fluorescent dyes	10	-	*Vibrio parahaemolyticus, Vibrio fluvialis, Yersinia enterocolitica, Proteus sp.,S. aureus, Enterococcus faecalis, Campylobacter jejuni, β-hemolytic Streptococcus, Listeria monocytogenes*	<2 h	[48]
*Salmonella and Campylobacter*	DNA Micro-array	14–57 and 11–60	-	*Listeria monocytogenes, B. cereus, Cronobacter sakazakii, Citrobacter freundii, Klebsiella pneumonia, E. coli, Proteus vulgaris, Enterobacter aerogenes, Hafnia alvei, Serratia marcescens*	45 min	[49]

^a^ LOD: the limit of detection cited in accordance with the IUPAC definition as “the smallest amount of concentration of analyte in the sample that can be reliably distinguished from zero”. CNP: Carbon nanoparticles; LAMP: Loop-mediated isothermal amplification; NASBA: Nucleic Acid Sequence Based Amplification; PMA: Propidium monoazide; RPA: Recombinase Polymerase Amplification; QD: Quantum Dots; q-PCR: quantitative Polymerase Chain Reaction.

**Table 2 sensors-20-05561-t002:** Selected examples of optical and related immunoassays for whole bacterial cells.

Strain/Analytical Scheme	Technique	LOD ^a^, CFU mL^−1^	Detection Range, CFU mL^−1^	Interference Studies	Assay Time	Ref.
*E. coli* O157:H7Immunomagnetic assay, separation from complex matrix	Plate counting method	16	1.6 × 10^1^–7.2 × 10^7^	*Salmonella enteritidis, Citrobacter freundii, Listeria monocytogenes*	15 min	[50]
*Staphylococus aureus* Aptaassay on microtiter plates	Colorimetric detection with AuNP as indicator	9	10–10^6^	*Vibrioparahemolyticus, Salmonella typhimurium, Streptococcus, E. coli, Enterobacter sakazakii, Listeria monocytogenes*	15 min	[51]
*Pseudomonas aeruginosa* Aptamer assay on MB	Fluorometric detection with magnetic separation	1	10–10^8^	*Listeria monocytogenes, S. aureus, Salmonella enterica, E. coli*	1.5 h	[52]
*Vibrio cholerae O1* Sandwich immunoassay	Chromatographic	5 × 10^5^–10^6^		*Shigella flexneri, Salmonella typhi, Pseudomonas aeruginosa, Proteus vulgaris, Klebsiella pneumonia, Enterobacter cloacae*	15 min	[53]
*E. coli* O157:H7 Sandwich immunoassay	ELISA with HRP-TMB label and AuNP for signal amplification	68 (PBS)	6.8 × 10^2^ (PBS) 6.8 × 10^3^ (in food)	*Salmonella senftenberg, Shigella sonnei, E. coli K12*	3 h	[54]
*Salmonella typhimurium* Apta- and immunoassay on MB	Colorimetric; ELISA on MB with HRP/TMB, and AuNP for signal amplification	1 × 10^3^	1 × 10^3^–1 × 10^8^	*Salmonella typhi, Salmonella paratyphi, S. aureus, E. coli*	3 h	[55]
*E. coli ATCC 8739* Apta-ssay on AuNP	FRET	3	5–10^6^	*E. coli DH5a, E. coli (ATCC 25922), Bacillus subtilis; S. aureus*	-	[56]
*Vibrio fischeri* Sandwich aptaassay on paper	Colorimetric detection with AuNP	40	40–4 × 10^5^	*Vibrio parahemolyticus, E. coli, Bacillus subtilis, Shigella sonnei, S. aureus, Salmonella choleraesuis, Listeria monocytogenes*	10 min	[57]
*E. coli* O157:H7 Sandwich immunomagnetic assay	Fluorescence using pH sensitive fluorophore release detection labels	15	-	*Streptococcus pneumoniae R6*	<3 h	[58]
*E. coli O157:H7, Salmonella typhimurium, Listeria monocytogens* Multiplex, Sandwich immunomagnetic assay	Fluorescence	<5	-	*No cross reactivity between target pathogens*	2 h	[59]
*Salmonella enterica* Sandwich and direct immunoassays	ELISA with CNT/HRP-TMB	10^3^ and 10^4^	-	-	24 h (direct); 3 h (sandwich)	[60]
*E. coli O157:H7, Salmonella typhimurium* Sandwich immunomagnetic assay	ELISA with HRP/TMB and AuNP network for signal amplification	3–15	-	*Listeria monocytogenes, Salmonella typhimurium, Salmonella enteritidis*	2 h	[61]
*Salmonella enterica typhi* Sandwich immunoassay with pre-enrichment in BPW	dot-ELISA, with Ab-HRP conjugate and 3,3 diaminobenzidine tetrahydrochloride	10^4^ before 10^2^ after enrichment	-	-	4 h, 10 h with enrichment	[62]
*Listeria monocytogenes, E. coli O157:H7 and Salmonella enterica* Sandwich immuno-fluorescence assay	Optical fiber; multiplexed simultaneous detection	10^3^	-	*Cross-reactivity tested with other target pathogens*	<24 h	[63]
*Escherichia coli* Lateral flow aptaassay on QD	Colorimetric	300–600	-	*Bacillus cereus, Enterococcus faecalis, Listeria monocytogenes, Salmonella enterica*	20 min	[64]
*Salmonella* Aptamer-based lateral flow assay	Colorimetric using up-conversion of NP for detection	85	150–2000	*E. coli, S. aureus, Bacillus subtilis*	30 min	[65]
*Salmonella typhimurium* Immunoagglutination-based immunoassay	Optical Mie scattering of antigen-Ab clusters	10 inconsistent with a 15 µL sample volume	100–10^6^	-	10 min (from 6 to 15 min)	[66]
*E. coli* O157:H7 Immunomagnetic pre-concentration	LRSP diffraction grated Au surface	50	10^3^ to 10^7^	*E. coli K12*	30 min	[67]
*Escherichia coli* Immunoassay in a paper-based microfluidic device	Optical Mie scattering of antigen-Ab clusters	10 inconsistent with a 3.5 µL sample volume	10 to 10^3^	-	90 s	[68]
*Salmonella typhimurium* Sandwich immunoassay with magnetic pre-concentration	Fluorescence detection using QDNPs	10^3^	10^3^–10^6^	*E. coli*	30 min	[40]
*RESONANCE-FREQUENCY-BASED IMMUNOASSAYS*						
*Escherichia coli* O157:H7 Immunoassay on Ab-modified glass b	Resonance frequency	1 (in PBS)	-	-	10 min	[69]
*Salmonella enterica* Aptamer-based assay on MB	Piezoelectric: QCM	100	100–4 × 10^4^	*E. coli*	40 min	[70]
*S. aureus* Aptamer-based assay	Magnetoelastic resonance frequency detection	5	10–1 × 10^11^	*Listeria monocytogenes, E. coli, Enterobacter sakazakii, Streptococcus, Vibrio parahemolyticus*	25–26 min	[71]
*Salmonella* Sandwich immunoassay	Piezoelectric: QCM using AuNP labels for mass amplification	10–20	10–10^5^	*Klebsiella pneumonia, Enterobacteria spp, Pseudomona spp, S. aureus*	9 min	[72]
*Listeria monocytogenes* Sandwich immunoassay o	Resonance frequency detection on a sputtered gold/ lead-zirconate-titanate surface	100	10^3–^10^5^	-	30 min	[73]

^a^ LOD: the limit of detection cited in accordance with the IUPAC definition as “the smallest amount of concentration of analyte in the sample that can be reliably distinguished from zero”. AuNPs: Gold Nanoparticles; BPW: Buffered Peptone Water; CNT: Carbon Nanotubes; ELISA: Enzyme-Linked Immunosorbent Assay; HRP: Horseradish Peroxidase; LRSP: Long Range Surface Plasmons; MB: Magnetic Beads; NP: Nanoparticles; TMB: Tetramethyl Benzidine; QCM: Quartz-Crystal Microbalance; QD: Quantum Dots; QDNPs: Quantum Dot Nanoparticles.

**Table 3 sensors-20-05561-t003:** Selected examples of ultrasensitive and/or specific sensors for bacterial cells based on electrochemical immunoassay approaches.

Strain/Analytical Scheme	Technique	LOD ^a^, CFU mL^−1^	Detection Range, CFU mL^−1^	Interference Studies	Assay Time	Ref.
*Sulphur reducing bacteria* Immunoassay on chitosan doped rGS	EIS at 10 mV vs. Ag/AgCl with ferricyanide	18	18–1.8 × 10^7^	*Vibrio angillarum*	-	[74]
*Sulphur reducing bacteria* Immunoassay on AuNP-modified Ni foam	EIS at 5 mV vs. Ag/AgC with ferricyanide	21	2.1 × 10^1^–2.1 × 10^7^	*Vibrio anguillarum, E. coli*	2 h	[75]
*Salmonella enterica* Immunoassay on gold electrodes	EIS at 5 mV vs.Ag/AgCl with ferricyanide	100 (10 CFU in 100 µL)	100–10 × 10^4^	*E. coli*	1.5 min (no data on incubation time)	[76]
*S. aureus (protein A)* Competitive magneto-immunoassay on TTF-AuSPE.	e-ELISA, HRP label; TTF mediator Amperometry at −0.15 V	1(raw milk)	1 to 10^7^	*E. coli, Salmonella choleraesuis*	2 h	[77]
*E. coli O157:H7, S. aureus* Immunoassay on nano-porous alumina	EIS at 25 mV vs. Pt; no label	100	-	*Both strains were used for specificity test*	2 h	[78]
*E. coli, Listeria monocytogenes, Campylo-bacter jejuni* Sandwich immunoassay with highly dispersed carbon particles	Electrochemical detection at 105 mV; HRP as a label, TMB as a substrate	50, 10 and 50, respectively	50–10^3^, 10–1500, and 50–500	-	30 min	[79]
*Pseudomonas aeruginosa* Aptaassay on AuNP and AuNP/SPCE	Amperometry at 0.4 V with TMB	60	60–60 × 10^7^	*Vibrio cholera, Listeria monocytogens, S. aureus*	10 min (colorimetry)	[80]
*E. coli* IDE modified with anti-*E. coli* Ab	Impedance at 5 mV: no label, electric field perturbation	300	10^2^–10^4^	-	1 h	[81]
*E. coli* O157:H7 Immunoassay at HA modified Au electrode	EIS with ferricyanide	7	10–10^5^	*S. aureus, Bacillus cereus, E. coli DH5a.*	-	[82]
*E. coli* O157:H7 Immunoassay on AuNP modified rGO paper	EIS at 5 mV vs. Ag/AgCl with ferricyanide	150	150–1.5 × 10^7^	*S. aureus, Listeria monocytogenes, E. coli DH5a.*	-	[83]
*E. coli* CIP 76.24 Immunoassay on polyclonal Ab/neuroavidin/SAM/Au	EIS with no indicator, at −0.6 V in aerated solutions	10	10–10^5^ and 10^3^–10^7^ for lysed cells	*S. epidermis:* Interference at ≥100 CFU mL^−1^	1 h incubation + detection	[84]
*E. coli* K12, MG1655 Phage typing & assaying activity of β-D-galactosidase in cell lysates, SPCE	Amperometry at 0.22 V, oxidation of enzymatically produced *p*-aminophenol	1 CFU in 100 mL	1–10^9^	*Klebsiella pneumoniae*	6–8 h	[85]
*E. coli* O157:H7 Immunoassay on monoclonal Ab/ITO	EIS at 0.25 V with ferricyanide as a redox indicator	10	10–10^6^	*S. typhimurium, E. coli K12*	0.8 h incubation + wash./detect.	[86]
*E. coli ORN 178* Assay at carbohydrate modified SAM on Au	EIS at 5 mV vs. Ag/AgCl; with ferricyanide	100	120 –2.5 × 10^3^	*E. coli ORN 208*	<1 h	[87]
*E. coli* XL1-Blue; K12 Assay on non-lytic M13 phage/AuNP/GCE	EIS with ferricyanide redox indicator, at 0.15 V	14	10–10^5^	*Pseudomonas chlororaphis*	0.5 h incubation + wash./detect.	[88]
*E. coli* O157:H7 Sandwich immunoassay on MB at Au IDE, detected a response to urea hydrolysis by urease	EIS at 0 V, no indicator, label: urease/AuNP/aptamer;	12	12–1.2 × 10^5^	*S. typhimurium, Listeria monocytogenes*	ca. 2 h	[89]
*E. coli K12 and DH5α* Sandwich immunoassay on MBs; on nitrocellulose modified Gr	Chronocoulometry at 0.3 V; no redox indicator; label: cellulase	1 (PBS), 2 (milk)	1–4 × 10^3^	*E. agglomerans, S. aureus, Salmonella enteretidis, B. subtilis, P. putida*	3 h	[90]
*E. coli* O157:H7 Sandwich immunoassay on nanoporous alumina membrane	EIS at 25 mV/Pt; no label	10	10^0^–10^4^	-	-	[91]
*E. coli* Aptaassay on ITO modified with photoelectrochemical non-metallic NM	Potentiometric detection at 0.15 V (cathodic) and −0.4 V (anodic) (ratiometric detection)	2.9	2.9–2.9 × 10^6^	-	12 h	[92]
*E. coli* O157:H7 Immunoassay on nanoporous alumina membrane	EIS; no label	10 (PBS) 83.7 (milk)	10–10^5^	*S. aureus, Bacillus cereus, E. coli DH5a.*	-	[93]
*E. coli* O157:H7 Aptaassay on a paper modified with graphene nanoplatinum composite	EIS with ferricyanide indicator at 100 mV	4	4–10^5^	-	12 min	[94]
*E. coli K12* Sandwich immunoassay	Amperometry at −0.35 V, HRP as a label; substrates: HQ/BQ andH_2_O_2_	55 (PBS) 100 (milk)	10^2^–10^8^	*Pseudomonas putida*	1 h	[95]
*E. coli* O157:H7 Sandwich immunoassay with PtNCs coupled to GOD	Cyclic voltammetry from −0.15 V to 0.65 V	15	32–3.2 × 10^6^	*Salmonella typhi, Shigella dysenteriae Shigella flexneri*	30 min	[96]
*E. coli* O157:H7 Immunoassay on a SAM modified gold electrode	EIS with ferricyanide at 0 V vs. Ag/AgCl	2	30–3 × 10^4^	*Salmonella typhimurium*	45 min	[97]
*E. coli* Sandwich immunoassay on AuNP-structured electrode in an automated microfluidic chip	Ammperometry at –0.1 V, with an HRP label and TMB as a substrate	50	50–10^6^	*Shigella, Salmonella spp., Salmonella typhimurium, S. aureus*	30 min	[98]
*E. coli O157:H7, S. aureus* Nano-porous alumina membrane in a PEG-modified microfluiidc chip	Electrochemical impedance	100	10^2^–10^5^	*E. coli O157:H7, S. aureus*	<1 h	[99]

^a^ LOD: the limit of detection cited in accordance with the IUPAC definition as “the smallest amount of concentration of analyte in the sample that can be reliably distinguished from zero”. AuNPs: Gold Nanoparticles; AuSPE: Gold Screen Printed Electrodes; BQ: Benzoquinone; EIS: Electrochemical Impedance Spectroscopy; GCE: Glassy Carbon Electrode; Gr: Spectroscopic Graphite; HRP: Horseradish Peroxidase; HA: Hyaluronic Acid; HQ: Hydroquinone; IDE: Interdigitated Electrodes; ITO: Indium Tin Oxide; GOD: Glucose Oxidase; MB: Magnetic Beads; NM: nanomaterial; PtNCs: Platinum Nanochains; PEG: Polyethylene Glycol; rGO: Reduced Graphene Oxide; rGS: Reduced Graphene Sheet; SAM: Self-Assembled Monolayers; SPE: screen printed electrodes; SPCE: Screen Printed Carbon Electrodes; TMB: 3,3′,5,5′-tetramethylbenzidin; TTF: TetraThiaFulvalene.

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
