# Peer review of "Electrochemical Immuno- and Aptamer-Based Assays for Bacteria: Pros and Cons over Traditional Detection Schemes"

_sensors, 2020, doi:10.3390/s20195561_

Round 1
Reviewer 1 Report
The manuscript pertains to an important theme that is timely and important. However, the authors need to revise it based on the provided comments below before it can be accepted for publication. 1. The title of the manuscript should be changed to "Electrochemical immunoassays for bacteria: A review". The current one is verbose and confusing. 2. There are many incidences in the text, such as Abstract, where the English Grammar has not been used correctly. So it needs to be improved by an expert. 3. Introduction should also include the role of various immunoassays in the detection of bacteria. Why the conventional immunoassays are not suitable and what is the emerging need for eletrochemical immunoassays? 4. Section 2 needs to include a table that clearly shows the bioanalytical characteristics for various detection techniques being used for bacterial detection. The text is verbose and doesnt convey the main points. 5. Fig 1 can be deleted and all the info can be summarized in the table above. 6. Section 3 should be merged with section 2 and if desired the authors can make subsections. 7. Section 4 is very confusing. Section 4.1 and 4.2 are both immunoassays only. So authors should merge them and if desired differentiate based on assay duration or application of immunoassays (i.e. central lab, point of care) or type (manual or automated), etc. 8. Section 4.4 and 4.5 are also confusing nomenclatures. Section 4.4 is electrochemiluminescence immunoassays. Section 4.5 should be merged into other section based on how authors wants to make sections. 9. The authors needs to include a section called "Emerging trends" just before conclusions. Here the authors should mention what approaches are being used internationally in making the bacterial detection automated, fast, integrated and POC. Several approaches have been used, which have not been mentioned by the authors that have far more outreach than the current ones mentioned in this manuscript. These include fully automated lab-on-a-chip based immunoassays, microfluidic immunoassays such as fully integrated centrifugal microfluidics platforms or vertical microfluidics, lab-in-a-tube. The clinical standard for immunoassays for 95% of analytes is automated chemiluminescent immunoassays based on random access analyzer that employs paramagnetic beads and detects analytes in less than 20 min.Author Response
Please see the attachment

Reviewer 2 Report
The review is well-written with detailed descriptions of electrochemical biosensors for bacteria detection. I would recommend a minor revision of the manuscript with the possible suggestions below:
- The review would be strengthened if the authors could include at least one example given with a figure to explain the sections of 4.3 Whole cell imprinted polymer sensors, 4.4 Electro-optical immunoassays and 4.5 Nanopore technology for whole cells e-immunoanalysis. There are no highlighted examples of these sections with figures currently. They can improve these sections with a recent and high-impact example for each section.
- The Table has to be carefully drawn as some of the information cannot be read well because the cells are are squeezed together. I would recommend the authors to remove some cells and organize the Table in a way that the text can be read clearly. I would recommend them to double-check the PDF file of this Table before submission carefully.
Reviewer 3 Report
The manuscript "Electrochemical Immuno- and Aptamer-Based Assays for Bacteria: Pros and Cons over Traditional Detection Schemes" is a review with good disscussion and it has several references cited. In the part of Perspectives there is a interesting discussion and I believe that the manuscript can be published in the journal.
Round 2
Reviewer 1 Report
The authors have replied to the comments and took some comments into consideration to improve their draft. The revised draft can be considered for publication.